# CD44 Depletion in Glioblastoma Cells Suppresses Growth and Stemness and Induces Senescence

**DOI:** 10.3390/cancers14153747

**Published:** 2022-07-31

**Authors:** Constantinos Kolliopoulos, Mohamad Moustafa Ali, Casimiro Castillejo-Lopez, Carl-Henrik Heldin, Paraskevi Heldin

**Affiliations:** 1Department of Medical Biochemistry and Microbiology, Uppsala University, P.O. Box 582, SE-751 23 Uppsala, Sweden; konstantinos.kolliopoulos@kemi.uu.se (C.K.); mohamad.ali@imbim.uu.se (M.M.A.); c-h.heldin@imbim.uu.se (C.-H.H.); 2Department of Genetics and Pathology, Uppsala University, P.O. Box 582, SE-751 23 Uppsala, Sweden; casimirocastill@gmail.com

**Keywords:** CD44, glioblastoma, RNA-sequence analysis, PDGF family members, hyaluronan biology, HAS2, stem cell-like enriched spheres

## Abstract

**Simple Summary:**

The hyaluronan receptor CD44 has an important role in glioblastoma multiforme (GBM) progression, but the precise mechanisms have not been elucidated. We have analyzed U251MG glioma cells, expressing CD44 or not, and grown in stem cell-like enriched spheres. Our results revealed that CD44 is important for cell growth and stemness, and for the prevention of senescence. Analysis by RNA sequencing revealed that CD44 is important for the interaction with the hyaluronan-enriched microenvironment. In addition, CD44 depletion impairs certain gene signatures, such as those for platelet-derived growth factor (PDGF) isoforms and PDGF receptors, as well as signatures related to hypoxia, glycolysis, and anti-tumor immune responses.

**Abstract:**

Glioblastoma multiforme (GBM) is a lethal brain tumor, characterized by enhanced proliferation and invasion, as well as increased vascularization and chemoresistance. The expression of the hyaluronan receptor CD44 has been shown to correlate with GBM progression and poor prognosis. Here, we sought to elucidate the molecular mechanisms by which CD44 promotes GBM progression by knocking out (KO) CD44, employing CRISPR/Cas9 gene editing in U251MG cells. CD44-depleted cells exhibited an impaired proliferation rate, as shown by the decreased cell numbers, decreased Ki67-positive cell nuclei, diminished phosphorylation of CREB, and increased levels of the cell cycle inhibitor p16 compared to control cells. Furthermore, the CD44 KO cells showed decreased stemness and increased senescence, which was manifested upon serum deprivation. In stem cell-like enriched spheres, RNA-sequencing analysis of U251MG cells revealed a CD44 dependence for gene signatures related to hypoxia, the glycolytic pathway, and G2 to M phase transition. Partially similar results were obtained when cells were treated with the γ-secretase inhibitor DAPT, which inhibits CD44 cleavage and therefore inhibits the release of the intracellular domain (ICD) of CD44, suggesting that certain transcriptional responses are dependent on CD44-ICD. Interestingly, the expression of molecules involved in hyaluronan synthesis, degradation, and interacting matrix proteins, as well as of platelet-derived growth factor (PDGF) isoforms and PDGF receptors, were also deregulated in CD44 KO cells. These results were confirmed by the knockdown of CD44 in another GBM cell line, U2990. Notably, downregulation of hyaluronan synthase 2 (HAS2) impaired the hypoxia-related genes and decreased the CD44 protein levels, suggesting a CD44/hyaluronan feedback circuit contributing to GBM progression.

## 1. Introduction

Glioblastoma multiforme (GBM) is the most aggressive and lethal type of brain tumors worldwide and has a poor prognosis. Its main attributes include high heterogeneity, elevated invasive and proliferative capacity, chemoresistance, increased vasculature, and a high rate of recurrence. So far, treatment with alkylating/methylating agents, such as temozolomide, has been the most widespread therapeutic approach together with radiotherapy, with modest clinical benefit [1]. Therefore, elucidation of the molecular mechanisms driving tumorigenic potential and invasiveness in GBM is of utmost importance for improving current therapeutic strategies.

CD44, a type I single-pass transmembrane glycoprotein, is an adhesion molecule and a well-established stem cell marker in various types of tumors. CD44 is encoded by a single gene and is generated by 20 exons in humans, giving rise to a plethora of isoforms due to alternative splicing. Post-translational modifications, such as extensive *O-* and *N-*glycosylation, add to the complexity of its biology. CD44 can be subjected to sequential proteolytic cleavage events, inducing ectodomain shedding and, after a second cleavage in the transmembrane domain, liberation of its intracellular domain (CD44-ICD), which is translocated to the nucleus and affects transcriptional responses [2]. Stimulation by growth factors and cytokines efficiently activate CD44 cleavage in various types of cancers, such as transforming growth factor β (TGFβ) in lung cancer [3] and platelet-derived growth factor (PDGF)-BB in breast cancer [4].

CD44 binds hyaluronan, a constituent of the extracellular matrix (ECM), via the Link domain in its N-terminus, which triggers downstream signaling by promoting the interaction of the CD44 cytoplasmic tail with members of the ERM family and ankyrin [5,6]. Hyaluronan–CD44 interactions have been shown to be pivotal in various aspects of tumor progression, such as proliferation, migration, invasion, survival, and stemness. During cancer progression, hyaluronan synthase 2 (HAS2) and the cell migration inducing hyaluronidase 2 (CEMIP2, also known as TMEM2), are upregulated, promoting tumorigenesis [7,8]. HAS2 expression is regulated both at the transcriptional and translational levels; for instance, in response to PDGF-BB and phorbol 12-myristate 13-acetate (PMA) [9,10,11,12], but also through the long non-coding RNA HAS2-AS1 [13] that is elevated in gliomas [14,15]. Hyaluronan is an abundant component of the brain extracellular matrix and promotes glioma cell invasion; however, the molecular mechanisms involved are not well understood [16,17].

CD44 levels correlate with GBM progression [18] and high levels of CD44 promote invasion of tumor cells [19]. Moreover, treatment with an anti-CD44 monoclonal antibody impeded glioma growth in mice [20]. The binding of CD44 to osteopontin, which is also abundant in brain tumors, leads to enhanced CD44 cleavage and the released CD44-ICD enhances the expression of hypoxia-inducible factor HIF-2α, promoting a hypoxic response in necrotic and perivascular areas [21].

The aim of this study was to elucidate the mechanisms involved in the pro-tumorigenic effects of CD44 in GBM. Depletion of CD44 in U251MG glioblastoma cells caused decreased proliferation and stemness, and acquisition of a pro-senescence state. Transcriptomic analysis of U251MG cells, grown in stem cell-enriched sphere-like conditions, revealed that CD44 is required for the expression of enzymes involved in the synthesis and degradation of hyaluronan, PDGF isoforms, and PDGF receptors, as well as for genes related to the reactive oxygen species (ROS) pathway, glycolysis, and progression from the G2 to M phase. Intriguingly, HAS2 knockdown led to decreased CD44 levels and impaired expression of hypoxia-related genes, implying a CD44/hyaluronan double-positive feedback circuit, driving GBM progression.

## 2. Materials and Methods

### 2.1. Cell Culture and Reagents

Human glioblastoma U251MG [22] and U2990 (kindly provided by Aristidis Moustakas, Uppsala University) [23] cells were grown in RPMI-1640 (Sigma-Aldrich Sweden AB, Stockholm, Sweden), supplemented with 10% fetal bovine serum (FBS) (Biowest, Biotech-IgG AB, Lund, Sweden), penicillin and streptomycin (100 μg/mL), and 5 mM L-glutamine (Sigma-Aldrich Sweden AB, Stockholm, Sweden). Cells were maintained at 5% CO_2_ in a humidified atmosphere at 37 °C. For non-adherent conditions, U251MG cells were grown for six days in ultra-low attachment 6-well plates (Corning, Sigma-Aldrich Sweden AB, Stockholm, Sweden). Culture medium was composed of DMEM/F12 (Gibco, Life Technologies Europe BV, Stockholm, Sweden), supplemented with B27 containing vitamin A (Invitrogen, Life Technologies Europe BV, Stockholm, Sweden), including also bFGF (20 ng/mL) and EGF (20 ng/mL) (PeproTech EC Ltd. Nordic, Stockholm, Sweden). The γ-secretase inhibitor DAPT (Sigma-Aldrich Sweden AB, Stockholm, Sweden) was used at the indicated concentrations; for low-attachment conditions, DAPT was added at final concentration of 2.5 μM every two days from the start of the experiment. The hyaluronan synthesis inhibitor 4-methylumbelliferone (4-MU, #M1508) was purchased from Sigma-Aldrich (Merck, Darmstadt, Germany) and used at a concentration of 0.5 mM from the start of the experiment. For evaluation of CD44 cleavage cells were treated with stem cell factor (SCF), PDGF-BB, tumor necrosis factor alpha (TNFα), TGFβ, hepatocyte growth factor (HGF) (PeproTech EC Ltd. Nordic, Stockholm, Sweden or Biosource Inc., Dacula, GA, United States). Final concentration of the factors was at 10 ng/mL.

### 2.2. Generation of CD44 KO Cells

*CD44* KO U251MG cells were established by CRISPR/Cas9 gene editing technology. Single-guide RNAs (sgRNA) were designed using the online tool at www.broadinstitute.org/gpp/public/analysis-tools/sgrna-design (accessed on 16 September 2016) and cloned into the BsmBI site of the lentiCRISPRv2 lentiviral vector [24]. Two sgRNA were designed to target two distinct sequences within exon 2 (166 bp in length) of the human gene according to the GTEx Portal (www.gtexportal.org, accessed on 16 September 2016). Exon 2 is the first constitutive exon, not subjected to alternative splicing. The sgRNA sequences were G60, 5′-CTTTGCAGGTGTATTCCACGtgg-3′, and G40, 5′-CTACAGCATCTCTCGGACGGagg-3′ (PAM sequences shown in lower case letters). Lentiviruses containing the sgRNA, the Cas9 nuclease, and puromycin N-acetyl-transferase genes were generated in HEK293T cells by co-transfection of the packaging plasmids psPAX.2 and psMD2 (Addgene, Watertown, USA). Supernatants containing lentiviruses were harvested 24 h and 48 h post-transfection. For efficient transduction, polybrene (8 μg/mL) (Sigma-Aldrich Sweden AB, Stockholm, Sweden) was added to the medium. Infected cells were selected in the presence of puromycin (0.5 μg/mL) (Gibco, Life Technologies Europe BV, Stockholm, Sweden). Thereafter, they were sparsely seeded (50, 100 and 250 cells) in 10 cm plates for each sgRNA for clone selection. After expansion of single clones, screening was performed via immunoblotting and confirmed KO clones were selected for experiments. Clone #13 was excluded for further analysis because of potentially expressing a truncated version of CD44.

### 2.3. Generation of shHAS2 KD Cells

Viral transduction of lentivirus shRNA particles targeting HAS2 (MISSION, cat. # SHCLNV), with pLKO.1-puro as host vector, were obtained from Sigma-Aldrich Sweden AB, Stockholm, Sweden. MISSION non-targeting mammalian shRNA control transduction particles (SHC002V) were used as control. Delivery was carried out according to the manufacturer’s instructions with a multiplicity of infection equal to 5 and use of polybrene (8 μg/mL) (Sigma-Aldrich Sweden AB, Stockholm, Sweden). Successfully transduced cells were selected in medium containing 1 μg/mL pyromycin (Calbiochem, Merck, Darmstadt, Germany).

Conditioned media from U251MG cells, grown in low-attachment conditions, or not, were collected and the hyaluronan concentration was quantified and normalized to 1 μg of total RNA extracted from the cells, as previously reported [3].

### 2.4. siRNA Transfection

siRNAs targeting CD44 or scrambled siRNAs (Invitrogen, Life Technologies Europe BV, Stockholm, Sweden) were transiently transfected to U2990 cells at 40 nM at two consecutive days. Following transfection, the cells were trypsinized and seeded at non-adherent conditions. Silentfect (Biorad Laboratories AB, Sundbyberg, Sweden) was used as a liposome reagent, according to the manufacturer’s instructions.

### 2.5. RNA-Sequencing and Gene Set Enrichment Analysis

Parental or CD44 KO (clone #28) U251MG cells were grown in low-attachment conditions for six days. Total RNA was extracted by employing ReliaPrep RNA Miniprep System kit (Promega, Madison, WI, USA). Sequencing libraries for six samples were prepared using a TruSeq stranded total RNA library preparation kit with RiboZero Gold treatment (Illumina, San Diego, CA, USA) following the manufacturer’s instructions. Sequencing was carried out on NovaSeq 6000 platform utilizing a whole lane of SP flow cell (Illumina) by the SNP&SEQ Technology Platform at the Science for Life Laboratory (Uppsala, Sweden). Three biological triplicates of each group were subjected to whole transcriptome analysis. The generated reads were mapped against the annotated human reference genome (GRCh38-hg38) and the annotated genecode primary assembly (genecode v38) using STAR aligner (v2.7.2b). The mapped reads were then quantified using the FeatureCounts tool of the Subread package (v2.0.0), and normalized to calculate the counts per million (CPM) values of each sample, followed by differential expression analysis using the DESeq2 package (v1.30.1) in R (v4.0.5). The gene set enrichment analysis (GSEA) was performed by utilizing the molecular signature database (MSigDB) and GSEA tool (v4.1.0) from the Broad institute. Gene sets with a nominal *p*-value < 0.05 and a false discovery rate (FDR) *q*-value < 0.1 were considered significant. Complete RNA-seq analysis can be found in Appendix A.

### 2.6. Extreme Limiting Dilution Assay (ELDA)

ELDA was carried out as previously described [25]. U251MG cells were grown in non-adherent conditions for 10 days in ultra-low attachment 96-well plates (Corning, Sigma-Aldrich Sweden AB, Stockholm, Sweden) after performing serial dilution ranging from 100 to one cell per well. After 10 days, micrographs of the formed spheres were captured by employing a Zeiss Axiovision 40 microscope (Carl Zeiss, Oberkochen, Germany) (5× magnification), and illustrative plots demonstrating the stem cell frequency for each condition were prepared by utilizing the online ELDA software program (http://bioinf.wehi.edu.au/software/elda, accessed on 8 December 2021). Data are plotted as the log fraction of wells without spheres as a function of the plated cell number. Spheres with diameter > 50 µm were scored under the inverted microscope.

### 2.7. Adhesion Assay

High molecular weight hyaluronan (HMW HA; 1000 MDa), kindly provided by Dr. Ove Wik (Q-Med, Uppsala, Sweden), was covalently immobilized on Covalink-NH microtiter plates (NUNC, Thermo Fischer Scientific, Gothenburg, Sweden) via mixing equal volumes of sulfo-NHS (0.184 mg/mL) (Thermo Fischer Scientific, Gothenburg, Sweden), containing HA (0.1 mg/mL) and EDC (0.123 mg/mL) (Sigma-Aldrich Sweden AB, Stockholm, Sweden) in each well. Fibronectin was kindly provided by Dr. Staffan Johansson, Uppsala University and used to coat plates at a concentration of 0.4 μg/mL. Collagen type I (PureCol, Advanced BioMatrix Inc., Carlsbad, CA, USA) was used at a concentration of 40 μg/mL. Plates were incubated overnight at 4 °C. The following day, the plates were washed with PBS and blocked with PBS containing 3% BSA (Sigma-Aldrich Sweden AB, Stockholm, Sweden) for 1 h at room temperature. Cells were starved overnight; the next day, cells were detached with PBS containing 10 mM EDTA, centrifuged, and resuspended in RPMI medium. Sixty thousand cells were incubated in the prepared 96-well plates in triplicate for 30 min at 37 °C, and then plates were washed carefully three times with PBS to discard non-attached cells. The attached cells were fixed and stained with 0.5% crystal violet in water containing 20% methanol for 20 min at room temperature, shaken in the dark, then washed thoroughly to remove excess dye, and dried completely. Finally, the bound dye was retrieved by adding pure methanol and incubating for 20 min at room temperature with shaking in the dark. Absorbance was measured at 570 nm.

### 2.8. Proliferation Assay

Parental or CD44 KO U251MG cells (100,000 per well) were seeded in 6-well plates in RPMI medium containing 5% FBS. Every three days, cells were trypsinized, counted by trypan blue staining, and reseeded in new 60-mm plates up until Day 6, when the cell number was counted.

### 2.9. Ki67 Immunofluorescence Microscopy

Immunofluorescence microscopy was carried out in parental or CD44 KO U251MG cells. After seeding, cells were starved for 24 h and subsequently cultured in RPMI medium, including 10% FBS, for 2 days. Then, cells were fixed for 10 min in 3.7% (*w*/*v*) formaldehyde in PBS, followed by permeabilization with 0.1% Triton X-100 in PBS for 10 min, blocked for 60 min with IgG-free 1% BSA in PBS, and incubated overnight at 4 °C with 0.4 μg/mL primary antibody against Ki67 (Santa Cruz Biotechnology, Dallas, USA, sc-23900) diluted in 1% BSA in PBS. Next day, coverslips were incubated with anti-mouse Alexa Fluor 546–conjugated secondary antibody (Invitrogen, Thermo Fisher Scientific) at a concentration of 1:1000 in 1% BSA in PBS for 1 h at room temperature in the dark. Extensive washes were performed between the aforementioned steps. Subsequently, coverslips were set onto glass slides and mounted by using 10 μL of VectaShield HardSet mounting medium containing 4′,6′-di-amidino-2-phenylindole (DAPI; Vector Laboratories, Burlingame, CA, USA) for nuclear visualization. A Zeiss Axioplan 2 fluorescence microscope was used with a Zeiss 20x objective lens. Images were acquired with a Hamamatsu C4742-95 CCD digital camera and the acquisition software QED Camera Plugin version 1.1.6 (QED Imaging Inc., Rockville, MD, USA) and Volocity 1 (PerkinElmer Life Sciences, Waltham, MA, USA). At least six random micrographs were retrieved per condition at the same exposure time and were quantified as a per cent of positive Ki67 nuclei using ImageJ software (v1.8.0_172).

### 2.10. Senescence-Associated β-Galactosidase (SA-β-Gal) Assay

The senescence-associated β-gal assay was performed as previously described [26]. Briefly, 200,000 parental or CD44 KO U251MG cells were starved overnight; the next day cells were treated with RPMI medium containing different serum concentrations ranging from 0 to 10% for 24 h. Cells were fixed in 2% formaldehyde and 0.2% glutaraldehyde in PBS for 5 min at room temperature. Following fixation, cells were incubated with staining solution containing 1 mg/mL 5-bromo-4-chloro-3-indolyl β-D-galactopyranoside (X-gal; Sigma-Aldrich Sweden AB, Stockholm, Sweden) for 12–16 h. Coverslips were washed with PBS, mounted, and observed under a Zeiss Axiovert 40 phase-contrast microscope to capture pictures of the cells with 10× objective lens. Bars represent 100 µm.

### 2.11. Immunoblotting

Cells were lysed in buffer containing 50 mM Tris-HCl, pH 8.0, 150 mM NaCl, 1% NP-40, 0.1% sodium dodecyl sulfate (SDS), 0.5% sodium deoxycholate, supplemented with HALT protease and a phosphatase-inhibitor cocktail (Thermo Fischer Scientific, Gothenburg, Sweden) and the protein concentration was measured by utilizing a BCA assay (Thermo Fischer Scientific, Gothenburg, Sweden). Samples with equal protein content were subjected to SDS-polyacrylamide gel electrophoresis (SDS-PAGE), followed by wet transfer to nitrocellulose membrane (Amersham, GE Healthcare, Uppsala, Sweden) and blocking in 5% non-fat milk in Tris-buffered saline (TBS), supplemented with 1% Tween 20. Subsequently, the membranes were incubated at 4  °C overnight with antibodies against CD44 (Abcam, Cambridge, UK), p16, tubulin (Santa Cruz Biotechnology), PDGFRA (homemade) [27], p-ERK1/2, ERK1/2, p-CREB, CREB, KIT, or GAPDH (Cell Signaling Technology, Leiden, The Netherlands), followed by incubation with horseradish peroxidase-conjugated secondary antibodies (1:10,000; Invitrogen, Life Technologies Europe BV, Stockholm, Sweden) for 1 h at room temperature, and development by chemiluminescence (Millipore, Burlington, MA, USA). Band intensity quantification was performed by using ImageJ software (v1.8.0_172).

### 2.12. RNA Extraction and Real-Time qPCR

RNA was extracted from cultures grown in spheres by using the RNeasy kit (QIAGEN AB, Sollentuna, Sweden) according to the manufacturer’s instructions. The iScript DNA synthesis kit (Biorad, Biorad Laboratories AB, Sundbyberg, Sweden) was used to reverse-transcribe 1 μg of total RNA to cDNA. KAPA Sybr Fast (PCR biosystems, Techtum Lab AB, Umeå, Sweden) was employed to perform real-time qPCR in triplicates (95  °C, 2 min; 40 × (95  °C, 10 s; 60  °C, 30 s)). The primers used for gene detection were the following: *KIT* forward, 5-CACCGAAGGAGGCACTTACACA-3 and reverse, 5-TGCCATTCACGAGCCTGTCGTA-3; *PDGFD* forward, 5-GTGGAGGAAATTGTGGCTGT-3 and reverse, 5-CGTTCATGGTGATCCAACTG-3; *HAS2-AS1* forward, 5-CAAGATCTTCTGAGGGGTGGAC-3 and reverse, 5-CTGAAAGGGATGGGTGAAAGGA-3; *HAS2* forward, 5-TCGCAACACGTAACGCAAT-3 and reverse, 5-ACTTCTCTTTTTCCACCCCATTT-3; *CEMIP2/TMEM2* forward, 5-GGAATAGGACTGACCTTTGCCAG-3 and reverse, 5-TTCTGACCACCCTGAAAGCCGT-3; *VCAN* forward, 5-GTGACTATGGCTGGCACAAATTCC-3 and reverse, 5-GGTTGGGTCTCCAATTCTCGTATTGC-3; *VEGFA* forward, 5-AGGAGGAGGGCAGAATCATCA-3 and reverse, 5-CTCGATTGGATGGCAGTAGCT-3; *LOX* forward, 5-ACTGCACACACACAGGGATTG-3 and reverse, 5-GCCTTCTAATACGGTGAAATTG-3; *CD44* forward, 5-ATAATAAAGGAGCAGCACTTCAGGA-3 and reverse 5-ATAATTGTGTCTTGGTCTCTGGTAGC-3; *PDGFB* forward, 5-TCCCGAGGAGCTTTATGAGA-3 and reverse 5-ACTGCACGTTGCGGTTGT-3; *PDGFRA* forward 5-TTCCTCTGCCTGACATTGAC-3 and reverse 5-GTCTTCAATGGTCTCGTCCTC-3; *BCL2* forward 5-AGGCTGGGATGCCTTTGTGG-3 and reverse 5-TTTGTTTGGGGCAGGCATGT-3; *HPRT* forward, 5-CCTGGCGTCGTGATTAGTGAT-3 and reverse, 5-AGACGTTCAGTCCTGTCCATAA-3; and *TBP* forward, 5-TGGCGTGTGAAGATAACCCAA-3 and reverse, 5-TCTTGGCAAACCAGAAACCCT-3. Gene expression was normalized to the housekeeping gene *TATA-Box Binding Protein* (*TBP)* or *Hypoxanthine Phosphoribosyl transferase (HPRT)*.

### 2.13. Statistical Analysis

Graphs show the mean ± SEM and are based on at least three independent experiments unless stated otherwise. Two-paired Student’s *t*-tests were used to calculate significance; three significance levels are indicated: * *p* < 0.05, ** *p* < 0.01, and *** *p* < 0.001.

## 3. Results

### 3.1. CD44 Depletion Diminishes the Proliferative Capacity and Induces Cellular Senescence in U251MG Cells

In an attempt to explore the role of CD44 in GBM progression, we utilized the CRISPR/Cas9 gene editing to generate stable CD44 knock-out (KO) clones of U251MG cells, a well-established, highly proliferative, and invasive GBM cell line. We designed two different single-guide RNAs (sgRNAs) targeting exon 2, one of the conserved exons encoding part of N-terminus of CD44 (Appendix A). Upon lentiviral delivery and puromycin selection, potential CD44 KO clones were screened by immunoblotting and Clones #1, #2, and #3, derived by using sgRNA #1, and Clones #20 and #28, derived by using sgRNA #2, were selected for further experiments (Figure 1a).

Since CD44 has been connected with cell cycle progression and proliferation of cancer cells of various origins, we analysed the proliferation of CD44 KO cells using a cell counting assay; we observed a significant reduction in cell numbers of KO clones compared to the parental cells (Figure 1b). To elucidate whether the decreased number of cells was due to impaired proliferative capacity or induction of apoptosis, immunostaining against the nuclear proliferation marker Ki67 was performed. We observed a reduced number of Ki67-positive nuclei in CD44 KO clones, such as Clones #3 and #28, compared to control cells, when cells were grown in a complete medium. In addition, parental cells deprived of serum also exhibited fewer Ki67-positive nuclei (Figure 1c,d). Furthermore, immunoblotting against cleaved caspase-3 was negative in CD44 KO clones, as well as in parental cells under these conditions (Kolliopoulos, C., unpublished data [3]), consistent with the notion that the reduction in the number of CD44 KO cells was due to impaired proliferation and not to induction of apoptosis.

Proliferation of glioma cells has been correlated to CREB activation [28], and since CD44 signalling is important for sustaining active CREB [29], we analysed the phosphorylation status of S132 in CREB by immunoblotting; dramatically reduced p-CREB levels were detected in CD44 KO clones in comparison to control cells, both in the presence and absence of serum (Figure 1e). Hence, these observations imply an essential role of CD44 in CREB activation and proliferation of U251MG cells.

Because the CD44 KO U251MG cells showed a reduced rate of proliferation, we investigated the possibility that they had undergone cellular senescence. For this purpose, we used a senescence-associated β-galactosidase (SA-β-gal) assay on cultures of the parental cells or selected CD44 KO clones grown in a medium without or with 5% serum. Interestingly, strong β-gal staining was detected in CD44 KO clones grown in the absence of serum compared to control cells; this effect disappeared by increasing the serum concentration, indicating that the abolition of CD44 signaling causes growth arrest and senescence under challenging culture conditions (Figure 1f). To further corroborate our observations, we monitored levels of the cell cycle inhibitor and well-established senescence marker p16/CDKN2A. Immunoblotting analysis revealed enhanced protein levels of p16 in CD44 KO clones compared to parental cells, an effect that was more prominent in the absence of serum (Figure 1g).

### 3.2. Genetic Depletion of CD44 Impairs GBM-Related Gene Signatures and Phenotypes in Sphere-Like Conditions

In order to address the contribution of CD44 to GBM progression, we performed RNA-sequencing of either U251MG KO cells (clone #28) or parental cells in the presence or absence of the γ-secretase inhibitor DAPT, which is known to inhibit CD44 cleavage; the CD44 cleavage product CD44-ICD has been implicated in mediating CD44-dependent transcriptional responses [30]. The cells were cultured in anchorage-independent sphere-like conditions. Our transcriptomic analysis revealed that, in total, 1684 genes were differentially expressed upon genetic depletion of CD44; 997 of them were down- and 687 upregulated in the CD44 KO cells compared to the control cells (Figure 2a and Appendix A and Appendix A). Moreover, treatment with DAPT led to altered expression of 1111 genes, of which 736 were down- and 375 were upregulated compared to control cells (Figure 2a and Appendix A and Appendix A). Importantly, 161 genes overlapped between CD44 KO cells and parental cells treated with DAPT (Figure 2a and Appendix A). Gene set enrichment analysis displayed significant downmodulation in gene signatures, characterizing, for example, hypoxia, the reactive oxygen species (ROS) pathway, the glycolytic pathway, and the G2-M phase cell cycle progression, both in CD44 KO cells and DAPT-treated cells, in comparison to parental cells (Figure 2b,c, Appendix A). By performing qPCR to validate the results of the RNA-seq assay, hypoxia-related genes VEGFA, LOX, and BCL2 mRNA were found to be downregulated in two CD44 KO clones (#28 and #3) and in parental cells treated with DAPT, compared to untreated parental cells (Figure 2d).

To evaluate the effect of CD44 depletion in stemness maintenance in sphere-like conditions, we performed an extreme limiting dilution assay (ELDA) in both parental cells and CD44 KO clones. Notably, all five CD44 KO clones exhibited diminished stem cell frequency compared to the control cells, stressing the importance of CD44 for self-renewal ability (Figure 2e).

### 3.3. CD44 Ablation in U251MG Cells Down-Regulates Expression of PDGF Family Members

To investigate the molecular mechanisms by which CD44 contributes to GBM progression, we focused on specific genes being affected by CD44 depletion in our transcriptomic analysis. The results of RNA sequencing revealed that the expression of members of the PDGF and PDGF receptor families, and particularly *KIT*, *PDGFRA*, *PDGFB*, and *PDGFD*, were significantly suppressed (Figure 3a). In particular, *KIT* was one of the most downregulated genes in the CD44 KO in comparison to parental cells (log2FoldChange −7.38, FDR = 1.31×10^−48^ (Figure 3a, Appendix A), and was found to be downregulated by DAPT treatment as well (log2FoldChange −1.87, FDR = 8.64×10^−30^) (Figure 3a, Appendix A). Downregulation of *KIT*, *PDGFRA*, *PDGFB*, and *PDGFD* mRNAs was validated by real-time qPCR analysis in CD44 KO Clones #28 and #3. Interestingly, the expression of *KIT,* but not of *PDGFRA*, was suppressed by treatment with DAPT, suggesting that KIT may be a CD44-ICD target gene. On the contrary, *PDGFB* and *PDGFD* were upregulated in the presence of DAPT (Figure 3b). These results were further corroborated at the protein level by immunoblotting for KIT and PDGFRA (Figure 3c). Intriguingly, screening for CD44 cleavage after treatment with several growth factors and cytokines revealed that the KIT ligand, stem cell factor (SCF), more efficiently induced shedding of the ectodomain of CD44 than ligands for other tyrosine kinase receptors (Appendix A).

### 3.4. CD44 Promotes the Synthesis of Hyaluronan in U251MG Cells and Their Adhesion to the Hyaluronan-Coated Substratum

Given that hyaluronan is a major component of the GBM stroma, we analyzed the impact of CD44 KO on hyaluronan synthesis and secretion (Figure 4). RNA-sequencing analysis revealed differences between CD44 KO and parental cells in the expression of mRNAs of molecules involved in hyaluronan synthesis, degradation, and its linkage to ECM proteoglycans. In particular, the *HAS2* levels dropped sharply (log2FoldChange −3.18, FDR = 8.9 × 10^−207^), as did the levels of its natural antisense transcript *HAS2-AS*1 (log2FoldChange −2.11, FDR = 4.45 × 10^−9^), which has been shown to be important for its expression [13]. Furthermore, *TMEM2*, encoding a cell surface-localized hyaluronidase, and *VCAN*, encoding the hyaladherin versican [31], were also decreased (log2FoldChange −1.68, FDR = 2.6 × 10^−101^ and log2FoldChange −2.05, FDR = 1.7 × 10^−168^, respectively) (Appendix A and Appendix A). These results were further validated by real-time qPCR of CD44 KO Clones #28 and #3 (Figure 4a). Interestingly, treatment with DAPT decreased the mRNA level of *HAS2-AS1* but increased those of *HAS2* and *VCAN*. Secreted hyaluronan levels in the conditioned medium were also reduced upon CD44 loss, but not after treatment with DAPT (Figure 4b).

In order to determine the binding capacity of U251MG cells to a hyaluronan-enriched substratum, we performed adhesion assays of CD44 KO cells or parental cells to wells on which hyaluronan was immobilized. Parental U251MG cells bound more avidly to hyaluronan-coated wells or wells which were coated with fibronectin and collagen type I, compared to uncoated wells. Interestingly, cells deprived of CD44 manifested decreased capacity to bind to uncoated wells, and their hyaluronan-binding ability was abrogated. Nonetheless, the binding of CD44 KO cells to fibronectin or collagen type I remained unaffected (Figure 4c).

In conclusion, our results support the notion that CD44 and CD44-ICD are important for the synthesis and degradation of hyaluronan, and for its interaction with ECM proteins. Moreover, CD44 is important for the expression of PDGF isoforms and PDGF receptors, which have been shown to promote hyaluronan synthesis [9,32].

### 3.5. HAS2 Knockdown Partially Inhibits CD44-Related Genes and Responses

Since KO of CD44 affected HAS2 expression concomitantly with hyaluronan synthesis, we sought to investigate whether downregulation of HAS2 would have an impact on CD44-mediated gene transcriptional responses and related phenotypes. By lentiviral delivery, two distinct shRNAs targeting *HAS2* (#3 and #4) were used to create stable shHAS2 U251MG knockdown cells. Interestingly, HAS2 knockdown decreased *VEGFA*, *LOX*, and *HAS2-AS1*, which have been correlated to hypoxia [33] (Figure 5a). Furthermore, we confirmed these results by transiently transfecting another GBM cell line, U2990, with a siRNA targeting CD44. Downregulation of *CD44* led to decreased expression of *HAS2*, *HAS2-AS1*, and *LOX* mRNAs, as was also observed in CD44 KO cells, further consolidating that expression of these genes depends on CD44 (Appendix A). We evaluated the self-renewal capacity of shHAS2 U251MG cells by performing ELDA and observed a reduction in stem cell frequency of HAS2 knockdown cells compared to control cells. In addition, treatment with the hyaluronan synthesis inhibitor 4-MU, or the γ-secretase inhibitor DAPT, dramatically diminished the stem cell numbers, further consolidating the importance of HAS2-mediated hyaluronan synthesis and CD44 cleavage, respectively, in self-renewal capacity (Figure 5b).

Interestingly, HAS2 knockdown led to a significant reduction in the phosphorylation of CREB and ERK1/2, as witnessed using two different shRNAs, phenocopying the result of CD44 genetic depletion, when cells were grown in sphere-like conditions (Appendix Aa,b). Additionally, HAS2 knockdown caused a decrease in the CD44 protein levels, implying a positive feedback loop mechanism involving CD44 and HAS2 in GBM (Appendix A).

## 4. Discussion

The aim of the present study was to shed light on the molecular events driving CD44-dependent GBM aggressiveness [34]. In line with the vast body of evidence implicating CD44 in glioma progression [16], we observed a significant reduction in the proliferative capacity of the well-established GBM in vitro model U251MG, upon depletion of CD44 by CRISPR/Cas9 gene editing. In the case of thyroid cancer cells, CD44 controls cell cycle progression and proliferation via sustaining phosphorylation and activation of the CREB transcription factor. CD44 cleavage and the liberated CD44-ICD seem to be required for this event [29]. Importantly, activation of CREB has been shown to be prominent and indispensable for proliferation of glioma tumors [28]. In agreement with these studies, we noticed a robust decrease in p-CREB levels in the presence or absence of serum in the CD44-deprived cells (Figure 1e), suggesting that activation of CREB downstream of CD44 is required to drive proliferation of U251MG cells.

Prolonged cell cycle arrest triggers senescence. We noticed that challenging cells by deprivation of serum in the culture medium caused upregulation of the cell cycle inhibitor p16/CDKN2A, promoting cellular senescence of CD44 KO cells. Notably, expression of CD44 has been inversely correlated with expression of p16, as well as p53, both having pivotal roles in senescence acquisition [35].

To delineate the role of CD44 in GBM progression, we performed RNA-seq analysis upon genetic depletion of CD44 or inhibition of its cleavage by the well-established γ-secretase inhibitor DAPT. Our analysis revealed a CD44-dependency for gene signatures related to hypoxia, glycolysis, ROS pathway, and progression from the G2 to M cell cycle phase. Interestingly, treatment with DAPT also affected similar pathways, indicating that at least a part of the CD44-dependent transcriptional responses is dependent on the generation of its cleavage product CD44-ICD. Our findings are consistent with a previous report showing that CD44 enhances the glycolytic pathway during cancer progression [36].

In cells grown in stem-cell enriching sphere-like conditions, the expression of molecules involved in synthesis and degradation of hyaluronan and members of the PDGF and PDGF receptor families were found to be dependent on CD44 expression. Importantly, *PDGFRA* (17.3%) and *KIT* (11.7%) have been found to be the most amplified genes encoding receptor tyrosine kinases (RTK) after *EGFR* (56%) in a cohort of more than 500 GBM tumors [37]. *PDGFRA* was strongly downregulated in CD44-depleted cells; concurrently, *PDGFB* and *PDGFD,* generating the homo-dimeric ligands PDGF-BB and PDGF-DD [38], were suppressed upon CD44 loss. Therefore, it is plausible that autocrine PDGF signaling via both PDGFRA and PDGFRB is impaired in U251MG cells upon CD44 KO. Our results also revealed that KIT expression was severely diminished upon CD44 loss or after inhibiting its cleavage, indicating that *KIT* may be a CD44-ICD target gene. Intriguingly, the KIT ligand SCF potently induced CD44 cleavage, suggesting a positive regulatory feedback loop between CD44 and SCF/KIT signaling. Noteworthy, we observed reduced stem cell frequency in CD44 KO cells compared to the parental U251MG, as shown by ELDA (Figure 2e). Our transcriptomic analysis revealed downmodulation of hypoxia-related genes in CD44 KO cells compared to the parental cells. It has been shown before that CD44-ICD stabilizes HIF2α, positively contributing to stemness [21]. Our results demonstrate partial inhibition of the hypoxia-related genes *VEGFA*, *LOX*, and *BCL2*, in CD44-depleted cells or when CD44 cleavage was abrogated by DAPT. The fact that *BCL2* expression is dependent on CD44 links CD44 biology to pro-survival mechanisms. Interestingly, in the inner mass of the sphere the cells need to bypass limited accessibility to nutrients and oxygen. Therefore, CD44^+^ cells with enhanced cleavage turnover would confer survival advantage compared to CD44-depleted cells or cells with decreased amount of CD44-ICD. According to our RNA-seq analysis and previous reports [39], CD44 positively regulates glycolysis and, interestingly, DAPT treatment suppresses glycolytic genes. Thus, CD44 and its cleavage product CD44-ICD might promote growth advantage characteristics by sustaining the metabolism of environmentally challenged cells.

Moreover, we observed an impairment in hyaluronan production and secretion in CD44 KO U251MG, grown in spheres, compared to the control cells, which may be attributed to HAS2 suppression (Figure 4a,b). Furthermore, *HAS2-AS1*, which is required for HAS2 expression, was also downregulated. The reduction in versican levels suggests global changes in the hyaluronan-formed matrix in sphere-like conditions (Figure 4a). Noteworthy, HAS2 knockdown diminished the CD44 protein levels as well as pCREB and pERK1/2 levels (Appendix A), concurrently with decreased hypoxia-related genes *VEGFA*, *LOX*, and *HAS2-AS1*, implying a potential role for HAS2 mediating CD44-transcriptional responses. Such double-positive feedback loops have been reported before in breast cancer [40].

In summary, we provide evidence that depletion of CD44 in U251MG cells leads to diminished cell growth and acquisition of senescence. Based on data from a transcriptomic analysis, we propose CD44-dependency for certain GBM hallmarks, such as hypoxia, enhanced glycolysis and cell cycle progression from the G2 to M phase. Our data support the notion that a CD44-HAS2 feedback circuit drives GBM progression.

## 5. Conclusions

GBM progression is promoted by HAS2-induced hyaluronan engaged in CD44 signaling and is closely related to the expressions of PDGF and PDGF receptor family members.

## Figures and Tables

**Figure 1 cancers-14-03747-f001:**
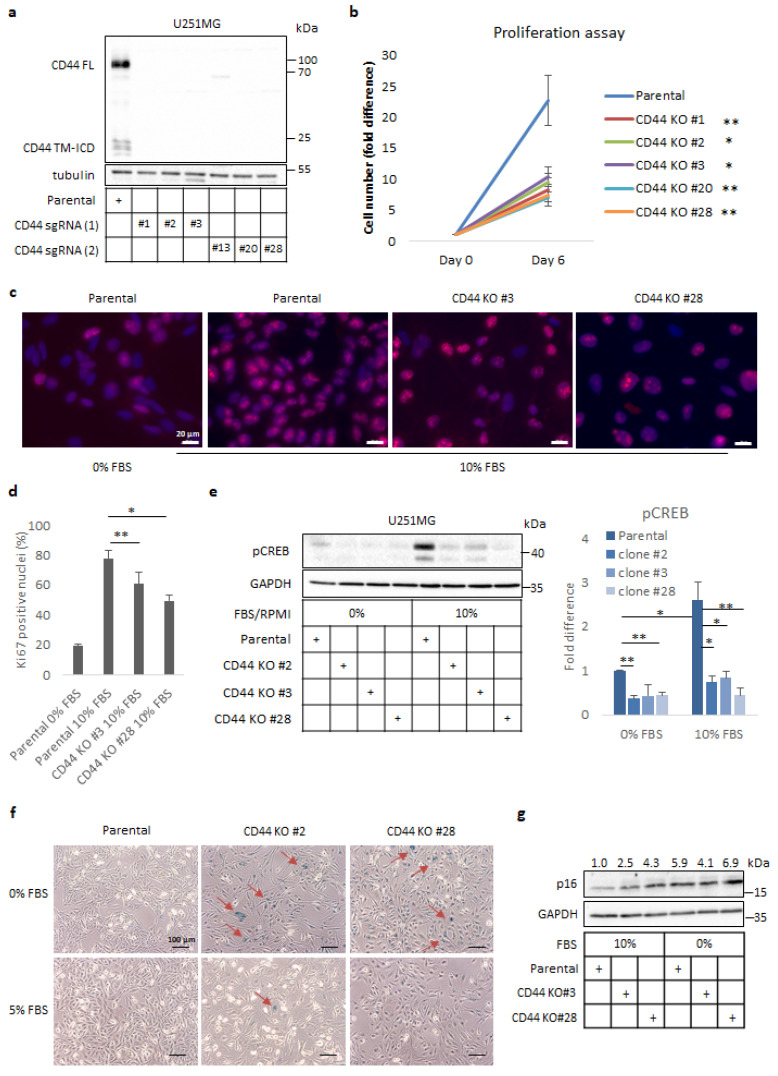
CD44 depletion reduces proliferation of U251MG cells and induces cellular senescence. (**a**) Immunoblot analysis of full length (FL) and cleavage products (TM-ICD) of CD44 in either parental or selected clones of CD44 KO clones of U251MG cells after utilizing two different single-guide RNA (sgRNA) constructs (1 and 2). Tubulin was used as the loading control. Uncropped immunoblots are shown in Appendix A. (**b**) Proliferation assay was performed by counting cells of parental U251MG and selected CD44 KO clones after 6 days in culture. (**c**,**d**) Immunofluorescence analysis of Ki67 expression in parental or CD44 KO U251MG cells cultured in 10% FBS containing medium for 24 h; culturing of parental U251MG cells in serum-free medium was used as the negative control (**c**). Quantification of Ki67-positive nuclei was carried out by using ImageJ software (**d**). Zeiss Axiovision was used to take the micrographs; bars, 20 μm. (**e**) Immunoblotting for phospho-(p)CREB after stimulation for 24 h in parental U251MG cells and CD44 KO clones cultured in media containing 10% or 0% FBS. GAPDH served as the loading control; the results are shown as the fold difference after normalization to GAPDH when cells were cultured in 10% FBS-containing medium. Uncropped immunoblots are shown in Appendix A. (**f**) A senescence-associated-β-gal assay was carried out in parental U251MG and selected CD44 KO clones after the cells had been cultured for 24 h in medium with 0 or 5% FBS. Zeiss Axiovision was utilized to take the micrographs. Bars represent 100 µm. (**g**) Immunoblotting analysis for p16 was performed following stimulation of cells for 24 h in medium containing 0% or 10% FBS; GAPDH was used as the loading control. Uncropped immunoblots are shown in Appendix A. Quantification of p16 after normalization to GAPDH is depicted above the immunoblots. All graph bars are illustrated as the average ± SEM based on at least three independent experiments. Asterisks depict significant differences compared to the control: * *p* < 0.05, ** *p* < 0.01.

**Figure 2 cancers-14-03747-f002:**
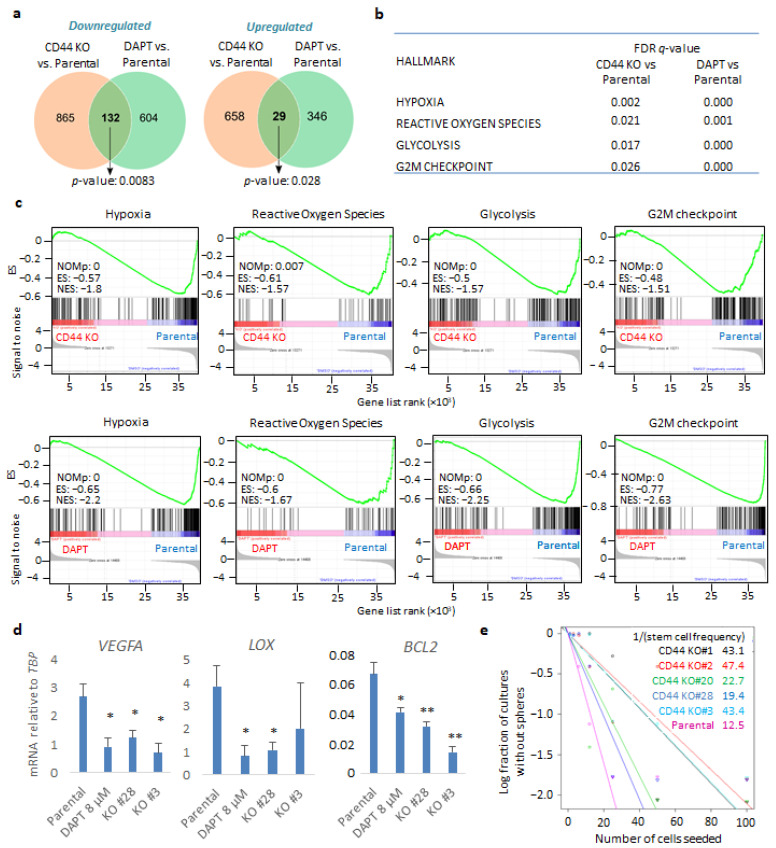
CD44 genetic depletion and inhibition of CD44 cleavage leads to common gene signatures in U251MG cells grown in sphere-like conditions. (**a**) Venn diagram of overlapping down- or upregulated genes comparing parental U251MG cells to either CD44 KO cells or U251MG cells treated with the γ-secretase inhibitor DAPT; *p*-values are computed using Fisher’s Exact test. Cells were grown in low-attachment conditions. (**b**) FDR *q*-values of specific gene signatures (hallmarks) upon CD44 KO or treatment of parental U251MG with DAPT, compared to control cells. (**c**) Specific gene set enrichment analyses of the transcriptome between parental U251MG cells and CD44 KO cells, or parental U251MG cells treated with DAPT. ES, enrichment score; NES, normalized ES; NOMp, nominal *p*-value. (**d**) Hypoxia-related genes VEGFA, LOX, and BCL2 mRNAs were quantified in the respective conditions by RT-PCR and demonstrated after normalization to TBP. Asterisks show significant differences compared to the respective control condition: * *p* < 0.05, ** *p* < 0.01. (**e**) Extreme limiting dilution assay (ELDA) was carried out of either parental U251MG or CD44 KO clones; stemness capacity was evaluated after cells had been grown under non-adherent conditions for ten days. Steeper slopes indicate higher stem cell frequency. A table demonstrating average stem cell frequency per group is shown. All graphs show the average ± SEM values from at least three independent experiments.

**Figure 3 cancers-14-03747-f003:**
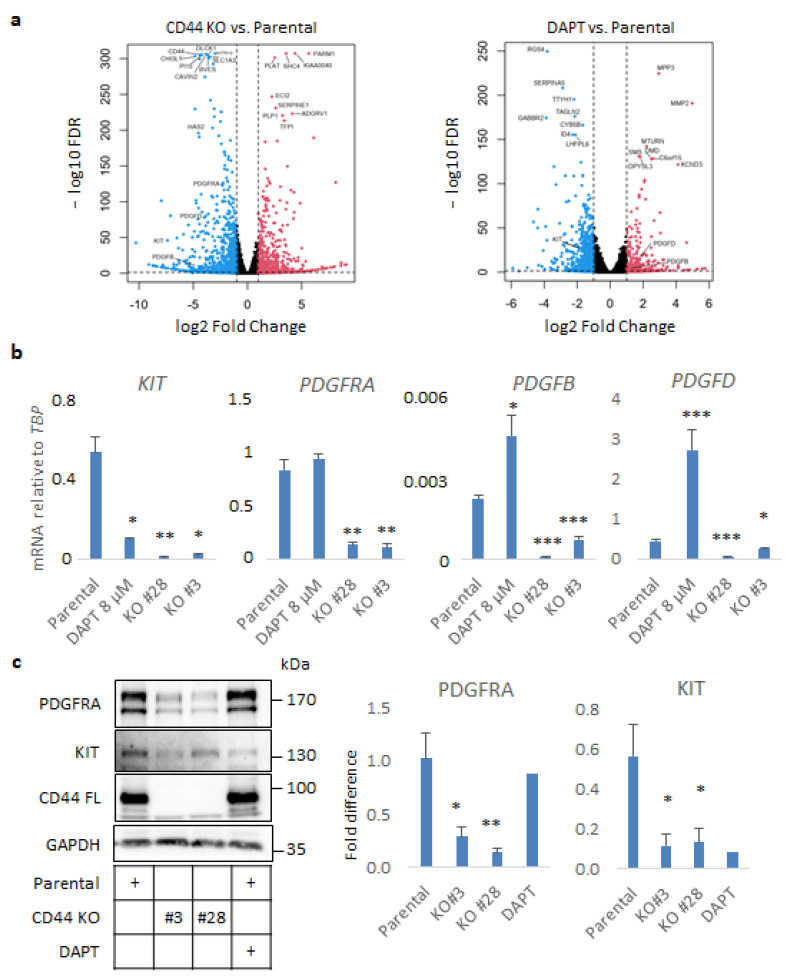
CD44 ablation impairs autocrine PDGF signaling in U251MG cells grown in sphere-like conditions. (**a**) Volcano plots exhibiting differentially regulated genes, which were either down- or upregulated (depicted with blue and red color, respectively) in CD44 KO cells or parental U251MG cells treated with the γ-secretase inhibitor DAPT, compared to the untreated cells. (**b**) mRNA levels of *KIT*, *PDGFRA*, *PDGFRB*, and *PDGFD* determined by RT-qPCR in spheres formed by parental U251MG cells treated or not with DAPT, and CD44 KO cells, are shown after normalization to *TBP.* (**c**) Immunoblotting analysis of PDGFRA, KIT, and CD44 was performed using lysates from U251MG cells. GAPDH was used as the loading control. Uncropped immunoblots are depicted in Appendix A. Immunoreactive bands were quantified after normalization to GAPDH after using the software ImageJ. Cells were grown in low-attachment conditions in the presence or absence of the γ-secretase inhibitor DAPT. All graphs illustrate the average ± SEM values from at least three independent experiments. Asterisks show significant differences compared to the respective control condition: * *p* < 0.05, ** *p* < 0.01, *** *p* < 0.001.

**Figure 4 cancers-14-03747-f004:**
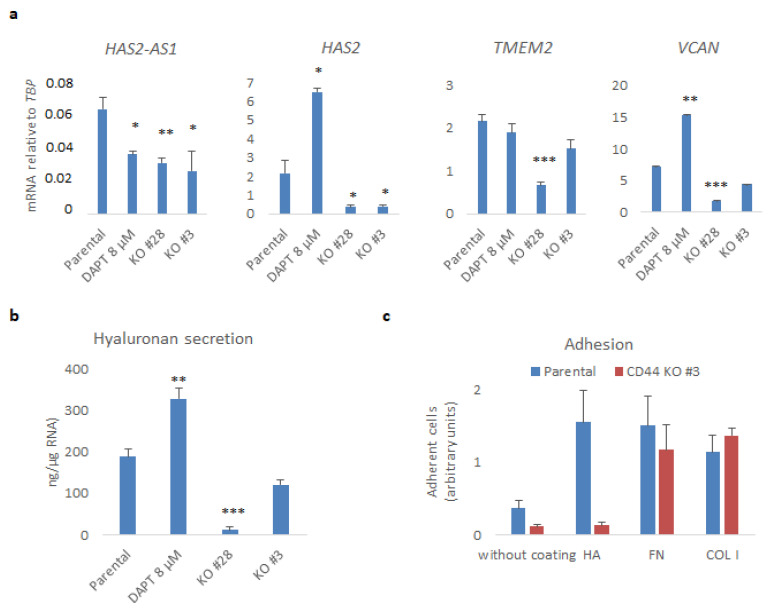
CD44 promotes hyaluronan production and adhesiveness of U251MG cells. (**a**) *HAS2-AS1*, *HAS2*, *TMEM2*, and *VCAN* mRNAs were determined by real-time qPCR analysis after normalization to *TBP*. (**b**) The hyaluronan levels in conditioned medium were measured and normalized to total RNA extracted from cells cultured in the indicated conditions. (**c**) An equal number of suspended parental or CD44 KO U251MG cells (Clone #3) was subjected to adhesion assay to uncoated plates or plates coated with hyaluronan (HA), fibronectin (FN), or collagen type I (COL I); after 30 min, adherent cells were stained with crystal violet. Results are depicted as arbitrary absorbance units. All graphs show the average ± SEM values of at least three independent experiments. Asterisks show significant differences compared to the respective control condition: * *p* < 0.05, ** *p* < 0.01, *** *p* < 0.001.

**Figure 5 cancers-14-03747-f005:**
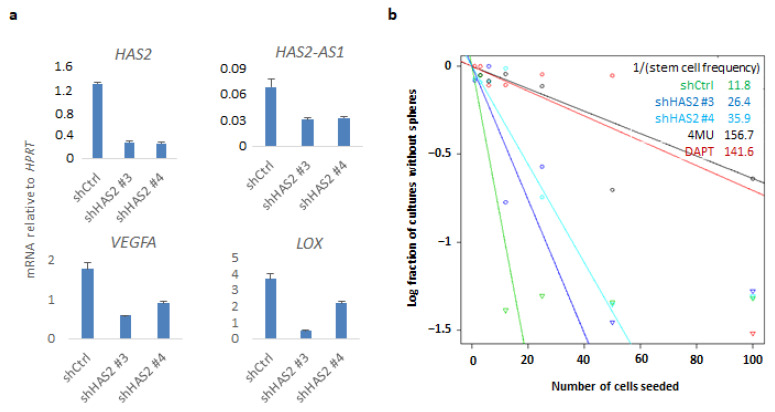
HAS2 downregulation impairs CD44-related genes and self-renewal capacity. (**a**) U251MG cells stably transfected with shRNAs against *HAS2* or control shRNA were grown in sphere-like conditions. *HAS2-AS1*, *HAS2*, *VEGFA*, and *LOX* mRNAs were quantified by real-time qPCR analysis and normalized to *HPRT*. (**b**) An ELDA assay was carried out in the *HAS2* knockdown and control cells, and cells treated with the hyaluronan synthesis inhibitor 4-MU or DAPT. One representative experiment out of two is depicted.

## Data Availability

The data presented in this study is available within the article or Appendix A.

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
