# Peer review of "CD44 Depletion in Glioblastoma Cells Suppresses Growth and Stemness and Induces Senescence"

_cancers, 2022, doi:10.3390/cancers14153747_

Round 1

Reviewer 1 Report

Konstantinos Kolliopoulos et al: CD44 Depletion in Glioblastoma Cells Suppresses Growth and Stemness and Induces Senescence

hit a very interesting topic of growth control for cancer treatment. There are some issues, which need to be addressed in the documentation of the scientific work and the data presented as per below:

Clarification as to why for certain experiments the 2D culture (attachment plates) and for others the 3D culture model (low attachment plates) was used

One would expect a more homogenous outcome for the 2D energy metabolism related signaling processes in 2D than in 3D because all cells have similar access to nutrients when cultured in 2D. 

I suppose the authors used 3D cultures for the RNA extraction and qPCR (please state in procedure described in 2.12). 

Discussion of Bcl2 downregulation (figure 2d) is missing, why was Bcl2 further analysed with PCR?

It is also not unlikely that there is a slightly different outcome for the caspase 3 activation and the proliferation assays when 2D and 3D cultures are compared. Does the change in Bcl2 expression play a role here?

The ELDA assay is applied multiple times, which criteria were used to classify a culture void of spheroids with one including spheres? Show this data in the main manuscript. See also M&M 2.6

M&M: 

2.1 please state if DAPT was present at the start of the culture.

2.2 probably state here why #13 was excluded in figure 1 and why most experiments were done with #3 and #28

2.6 what was the limiting size for a cell object to be classified as sphere, please show example micrographs with ELDA plots

When presenting microscopic imaging data please state the magnification and the NA of the objective used.

Figure 1: 

a: display full range of intensity down to background in CD44 blot image, little signal is still seen in lane CD44sgRNA2 #13 but this is probably an unspecific band?

b: why is clone #13 not included in the proliferation assay?

c: use cyan and magenta colors to allow for visualisation of nuclei and also Ki67, remove scale bar size in image and state size of scale bar in figure legend (writing is too small)

d: how was the analysis in d) performed (threshold for Ki67? imagej cell counter tool? Analyse\analyse objects tool? from how many samples and how many fields of view?)

e: also show pCREB difference in 0% FBS medium in same graph, indicate FBS content in legend of graph.

f: Brightness seems different in the different images it is therefore unclear how conclusions can be drawn about senescence. Indicate what the outcome is for the Beta-Gal assay e.g. put arrows into micrographs to show difference, quantify blue stain. Write size of object presented in the image in figure legend (scale bar).

g: The quantification of p16 relative to GAPDH as stated in the legend is not in the figure.

Figure 2:

please state why #28 for the KO was selected or if there was no reason (and results 3.2). was PCR performed in all clones?

state if the regulation of genes could be because of the localisation of the cell in the 3D cell culture re accessibility of nutrients and O2. 

Please compare spheroid size under different drug treatment conditions e.g. show 2D images of spheres, if that data is still available the authors, discuss

d: the large error in the Lox signal for #3 might be due to the normalisation or due to the Lox itself, please discuss briefly.e: please state why #13 was excluded from further analysis (see above)?

Please state if the localisation of the CD44 positive in cells grown in spheres could be an issue especially with the DAPT treatment, present data or discuss. 

Figure 3:

a: Vulcano plots: label CD44KO #? and DAPT treatment vs control?

c: there is no visible error bar for DAPT treatment for PDGFRA and KIT.

Figure 4: 

b: Was the amount of RNA/culture largely different or did indeed the reduced  Hyaluronan secretion cause the significant reduction in #28, please indicate p value for KO#3 vs parental.

c: adhesion is strongly affected in Parental vs Hyaluronan coated surfaces only, therefore the presence of FN or Col could abolish the effect of CD44KO on adhesion, 

how could this interfere with a pivotal role of CD44 for GBM treatment? Does CD44 play a role in sphere size?

Figure 5

a: please show WB data of shHAS (in supplement?)

Author Response

Reviewer#1

- Clarification as to why for certain experiments the 2D culture (attachment plates) and for others the 3D culture model (low attachment plates) was used

Pertinent to the Referee’s question about why both 2D and 3D cultures were used, both systems were employed depending on the assay. Low attachment plates and stem cell-like medium were used to address stemness as readout. For adhesion assay, single cells were in suspension, so they are devoid of physical interaction with each other or the culture dish. For β-gal staining, cells need to be in 2D. The proliferation assay was performed in 2D as starting the assay. We agree with the referee’s point about a more homogenous outcome for the 2D energy metabolism-related signaling compared to 3D condition, because of access to nutrients. Nonetheless, the 3D condition is indispensable for evaluating stemness potential.

- I suppose the authors used 3D cultures for the RNA extraction and qPCR (please state in procedure described in 2.12).

As the referee points out, RNA extraction and subsequent qPCR were performed from cultures grown in spheres. This has now been added to the text.

- Discussion of Bcl2 downregulation (figure 2d) is missing, why was Bcl2 further analyzed with PCR?

Bcl2 was further analyzed and validated because it was a strong hit in the hypoxia gene signature in our RNA-seq analysis. BCL2 downregulation in CD44 KO cells ranked higher than VEGFA or LOX, all well-established hypoxia-related genes (Table S3). There have been reports also linking CD44 to Bcl2 expression in multiple cell lines.

- It is also not unlikely that there is a slightly different outcome for the caspase 3 activation and the proliferation assays when 2D and 3D cultures are compared. Does the change in Bcl2 expression play a role here?

Regarding the role of Bcl2 in 2D and 3D and different outcomes of caspase 3, we haven’t analyzed the levels of Bcl2 in 2D conditions.

- The ELDA assay is applied multiple times, which criteria were used to classify a culture void of spheroids with one including spheres? Show this data in the main manuscript. See also M&M 2.6

For ELDA assays, the cutoff threshold for considering only well-formed spheres was well-formed spheres with diameters larger than 50 µm. An inverted microscope was used to capture the micrographs at 5x magnification. This has now been clarified in the text. Please, check the attached PPT.

M&M:

- 2.1 please state if DAPT was present at the start of the culture.

Both inhibitors DAPT and 4-MU were added from the start of the experiment. This has now been clarified in the text.

- 2.2 probably states here why #13 was excluded in figure 1 and why most experiments were done with #3 and #28

Clone#13 was excluded for further analysis because it may possibly express a truncated CD44. Please see the attached file. The band migrating a bit faster than CD44 full length (CD44 FL) was not observed in the other clones. Therefore, this clone was excluded and not further tested. Experiments were performed with clones #3 and #28 to have at least one clone representing different gRNAs and thus decreasing off-target related phenotypes.

- 2.6 what was the limiting size for a cell object to be classified as a sphere, please show example micrographs with ELDA plots

Please see our answer above.

- When presenting microscopic imaging data please state the magnification and the NA of the objective used.

NA of the used objective has now been stated.

Figure 1:

- a: display the full range of intensity down to the background in the CD44 blot image, the little signal is still seen in lane CD44sgRNA2 #13 but this is probably an unspecific band?

Please see the attached file. Probably a truncated protein, since it is not seen in the other samples. Excluded and not further tested.

- b: why is clone #13 not included in the proliferation assay?

Please see our answer above.

- c: use cyan and magenta colors to allow for visualization of nuclei and also Ki67, remove scale bar size in image and state size of scale bar in figure legend (writing is too small)

Distinct nuclei are now visible in the pictures depicting Ki67 staining.

- d: how was the analysis in d) performed (threshold for Ki67? ImageJ cell counter tool? Analyse\analyse objects tool? from how many samples and how many fields of view?)

Details have been added accordingly to the Materials and Methods section.

- e: also show pCREB difference in 0% FBS medium in the same graph, indicate FBS content in the legend of graph.

Quantification for the condition in the absence of serum has been added to the graph.

- f: Brightness seems different in the different images it is therefore unclear how conclusions can be drawn about senescence. Indicate what the outcome is for the Beta-Gal assay e.g. put arrows into micrographs to show the difference, and quantify the blue stain. Write the size of the object presented in the image in the figure legend (scale bar).

Arrows have been inserted to illustrate senescent cells (β-gal positive cells). Please find enclosed in the PPT the same micrographs enlarged. Cells also look larger, a hallmark of senescence phenotype.

- g: The quantification of p16 relative to GAPDH as stated in the legend is not in the figure.

Quantification of p16 immunoblot relative to GAPDH has now been given on top of the immunoblot.

Figure 2:

- please state why #28 for the KO was selected or if there was no reason (and results in 3.2). was PCR performed in all clones?

For RNA-seq, KO#28 was selected, since it produced the most reproducible results compared to the other clones in the assays used in the manuscript and experiments not shown. We further validated all assays with at least a second clone of the same or different gRNA. qPCR was not performed in individual clones for CD44 levels. Immunoblotting was used instead.

- state if the regulation of genes could be because of the localization of the cell in the 3D cell culture re accessibility of nutrients and O2.

Cells were grown in normoxic conditions (21% O2) and the cells at the outer layer of the sphere were exposed more to nutrients, nonetheless, the pool of cells was collected at the end, so the final result reflects the whole population. Another important parameter is the anchorage-independent growth in 3D. ECM architecture and composition are different and recapitulate better the physiological condition than the 2D cell culture.

- Please compare spheroid size under different drug treatment conditions e.g. show 2D images of spheres, if that data is still available the authors, discuss

There was a profound effect on sphere size as observed in representative micrographs between treatments with 4-MU or DAPT and control cells and also between HAS2 KD and control cells. The bar represents 100 µm.

- d: the large error in the Lox signal for #3 might be due to the normalization or due to the Lox itself, please discuss briefly.e: please state why #13 was excluded from further analysis (see above)?

LOX levels were fluctuating more from experiment to experiment for clone#3. Maybe this gene is more sensitive to minor external factors that are not easily controlled. The cells were cultured for 6 days. Small alterations in pH, nutrient deficiency, or aggregation of spheres could lead to such discrepancies.

- Please state if the localization of the CD44 positive in cells grown in spheres could be an issue, especially with the DAPT treatment, present data, or discuss.

CD44 positively affects hypoxia gene signature according to our findings. DAPT treatment abrogates CD44 cleavage and decreases hypoxia-related genes. Therefore CD44+ cells with enhanced cleavage turnover, inside the core of the sphere with limited accessibility to nutrients and oxygen, would confer a survival advantage compared to CD44-depleted cells or with a decreased pool of CD44-ICD. Low pH and lack of nutrients promote enhanced glycolytic flux (Warburg effect). According to our RNA-seq analysis, CD44 positively regulates glycolysis and interestingly DAPT treatment suppresses glycolytic genes. Thus, CD44 and its cleavage product CD44-ICD promote growth advantage characteristics by sustaining the metabolism of environmentally-challenged cells. This has now been discussed in the Discussion section.

Figure 3:

- a: Vulcano plots: label CD44KO #? and DAPT treatment vs control?

Labels were added to Fig 3a.

- c: there is no visible error bar for DAPT treatment for PDGFRA and KIT.

There is no error bar in the immunoblot including the DAPT treatment sample since one experiment was performed for DAPT treatment for immunoblotting to confirm qPCR results. Immunoblotting for KO vs parental was performed at least three times though.

 Figure 4:

- b: Was the amount of RNA/culture largely different or did indeed the reduced  Hyaluronan secretion cause the significant reduction in #28, please indicate the p-value for KO#3 vs parental.

Hyaluronan production differs between clone #28 and clone #3 even though HAS2 levels seem to be adequately downregulated in both clones. One possible explanation would be the differential expression of other HAS enzymes among different clones of CD44 KO as a compensatory mechanism for loss of hyaluronan production.

- c: adhesion is strongly affected in Parental vs Hyaluronan coated surfaces only, therefore the presence of FN or Col could abolish the effect of CD44KO on adhesion, how could this interfere with a pivotal role of CD44 for GBM treatment?

Hyaluronan is enriched in the brain and in abundance compared to collagen. Fibronectin is often upregulated in highly aggressive tumors. Double targeting of both Fibronectin and CD44 signaling may significantly suppress pro-survival signals and is a possibility worth investigating in the future.

- Does CD44 play a role in sphere size?

We have shown before that Has2-mediated hyaluronan production enhances sphere formation in breast cancer cells, increasing sphere size. Related to our finding that CD44 is the predominant adhesion molecule for hyaluronan in U251MG cells, mediating its downstream effects, it is plausible to speculate that CD44 would positively affect sphere growth via hyaluronan binding. This is further corroborated in our ELDA assays.

Figure 5

- a: please show WB data of shHAS (in supplement?)

Unfortunately, there are no known well-established antibodies for HAS2 to use in immunoblotting.  Usually, loss of silencing of Has2 is shown at the mRNA level.

Reviewer 2 Report

In this manuscript, authors demonstrate CD44 promotes GBM progression by HAS2-induced hyaluronan engaged CD44 signaling.  CD44 KO results in decreased stemness and increased senescence, impairs GBM-related gene signatures and phenotypes by RNA seq. Further studies also show HAS2 knockdown inhibits CD44-related genes and responses, suggesting a CD44 / hyaluronan feedback circuit contributing to GBM progression. Overall, it’s a good paper.

Here is concern:

1.       In vivo studies are recommended using CD44KO GBM cell implantation GBM model, whether CD44 KO results in glioma progression/angiogenesis/survival elongation?

2.       Authors used U251GBM cell line in this paper. Whether the results from U251 cell lines can mirror the primary GBM cell lines?

3.       Whether HAS2 Knockdown results in the same effect as CD44 KO (stemness, senescence)?

4.       typos in the manuscript. Delete the additional “, “on line 491.

Author Response

Reviewer#2:

  1. In vivo studies are recommended using the CD44KO GBM cell implantation GBM model, whether CD44 KO results in glioma progression/angiogenesis/survival elongation?

We agree that this would be interesting, however, this is beyond the scope of the current study.

  1. The authors used a U251GBM cell line in this paper. Whether the results from U251 cell lines can mirror the primary GBM cell lines?

U251MG is a well-established GBM cell line, of astrocytic origin, expressing high levels of GFAP and EGFR. Indeed, GBM is highly heterogeneous, composed of different subtypes. Nonetheless, we were able to reproduce our results in another GBM cell line.

  1. Whether HAS2 Knockdown results in the same effect as CD44 KO (stemness, senescence)?

HAS2 KD affects stemness as shown with the ELDA and sphere size. We have previously shown that HAS2 mediates cell cycle progression in breast cancer cells. Therefore, it is possible that HAS2 abrogation would lead to sustained cell cycle arrest and potentially senescence also in GBM cells. This is an interesting idea worth exploring in the future.

  1. typos in the manuscript. Delete the additional “, “on line 491.

The extra comma has been deleted.
